# Inelastic Deformable Image Registration (i-DIR): Capturing Sliding Motion through Automatic Detection of Discontinuities

Carlos I. Andrade [1,2,†] and Daniel E. Hurtado [1,2,3,*,†]

1 Department of Structural and Geotechnical Engineering, School of Engineering, Pontificia Universidad Católica de Chile, Santiago 7820436, Chile; ciandrade@uc.cl
2 Institute for Biological and Medical Engineering, Schools of Engineering, Medicine and Biological Sciences, Pontificia Universidad Católica de Chile, Santiago 7820436, Chile
3 Millennium Nucleus for Cardiovascular Magnetic Resonance, Santiago 7820436, Chile
* Correspondence: dhurtado@ing.puc.cl; Tel.: +56-2-23544207
† Current address: Avda. Vicuña Mackenna 4860, Macul, Santiago 7820436, Chile.

**Abstract:** Deformable image registration (DIR) is an image-analysis method with a broad range of applications in biomedical sciences. Current applications of DIR on computed-tomography (CT) images of the lung and other organs under deformation suffer from large errors and artifacts due to the inability of standard DIR methods to capture sliding between interfaces, as standard transformation models cannot adequately handle discontinuities. In this work, we aim at creating a novel inelastic deformable image registration (i-DIR) method that automatically detects sliding surfaces and that is capable of handling sliding discontinuous motion. Our method relies on the introduction of an inelastic regularization term in the DIR formulation, where sliding is characterized as an inelastic shear strain. We validate the i-DIR by studying synthetic image datasets with strong sliding motion, and compare its results against two other elastic DIR formulations using landmark analysis. Further, we demonstrate the applicability of the i-DIR method to medical CT images by registering lung CT images. Our results show that the i-DIR method delivers accurate estimates of a local lung strain that are similar to fields reported in the literature, and that do not exhibit spurious oscillatory patterns typically observed in elastic DIR methods. We conclude that the i-DIR method automatically locates regions of sliding that arise in the dorsal pleural cavity, delivering significantly smaller errors than traditional elastic DIR methods.

**Keywords:** deformable image registration; tissue sliding; lung biomechanics





## 1. Introduction

Deformable image registration (DIR) is an image-analysis technique used to determine the optimal transformation that establishes the spatial correspondence of a point between two images. When constructing a DIR method, three key elements need to be defined: (i) the transformation model, (ii) the regularizer, and (iii) the similarity measure [1]. These elements allow for the classification of DIR methods, and the reader is referred to [2] for a complete review. In particular, in this work we are concerned with the ability of the method to capture large displacements in the optimal transformation between medical images. From this perspective, transformation models can be divided into continuous-displacement transformations [3], which are suitable for small-deformation problems, and incremental diffeomorphic transformations based on the integration of flow equations [4,5], which can capture large deformations in DIR problems. While diffeomorphic methods have proven advantageous in capturing the large-displacement kinematics in DIR, continuous displacement models have been preferred in the field of medical imaging, as they provide a simple and efficient computational framework to DIR [6].

DIR has essential applications in radiology, such as the fusion of an anatomical image with a functional image [7], image-guided radiotherapy [8], and in treatment and surgery

planning [9]. DIR has proven fundamental in the study of the deformation mechanisms that take place at a regional level in human lungs, where the primary inputs for determining regional deformation are deformation measures based on the Jacobian matrix of the optimal transformation resulting from DIR of lung computed-tomography (CT) images [6,10]. DIR-based biomechanical analysis has revealed significant spatial differences in the magnitude, anisotropy, and heterogeneity of regional deformation in the lung of normal human subjects [10,11], measured in terms of volumetric change expressed either as a Jacobian determinant [12] or as regional volumetric strain and deformation invariants that quantify linear and surface changes [13]. Further, spatial patterns of regional deformation obtained from DIR have been found to significantly differ from normal lungs in asthmatic patients [14] and patients with chronic obstructive pulmonary disease [15], highlighting the potential of biomechanical analysis in understanding, and potentially detecting disease progression. Estimates of volumetric strain have been correlated with lung inflammation and injury in mechanically-ventilated lungs, suggesting that regional deformation obtained from DIR can be useful in the prevention of ventilation-induced lung injury in critical-care patients [16–18]. A fundamental limitation of current DIR techniques and libraries is the poor performance experienced when images display motion discontinuities such as contact and organ sliding. Sliding typically occur inside the human body due to the existing lubricated interfaces between internal structures in the thoracic cage (e.g., lungs, chest wall, heart) [10,19,20]; and between the liver and other abdominal organs (e.g., kidney, diaphragm) [21,22]. Interestingly, sliding in the lung fissures has been detected from computed-tomography (CT) images of the lungs using DIR methods, where supraphysiological levels of shearing deformation colocalize with the fissures [23]. While useful for anatomy detection purposes, no regional tissue distortion is expected to occur in sliding regions, which invalidates the accuracy of regional deformation estimates from traditional DIR methods in the lungs in regions close to fissures and the pleural cavity. The main responsible for such spurious deformation levels in sliding is the transformation model that most DIR methods assume, typically constructed using interpolation schemes that deliver globally continuous and smooth transformation mappings [24]. As a consequence, traditional DIR methods cannot capture material interface or motion discontinuities, thus hindering the accuracy of the image registration and the associated biomechanical analysis [11].

Traditional DIR techniques have been modified to capture sliding either by using alternative regularization terms, as well as enhanced transformation models [24–27]. One example of the former is the diffusion-based approach [28], where the normal component of the displacement field near the sliding boundaries is continuous, and a direction-dependent regularization term is assumed such that it penalizes jumps in the normal direction but allows for a discontinuous displacement field in the tangential direction [25]. This direction-dependent registration model shows good registration accuracy but underperforms when the intensity contrast near the boundaries is low, which can be the case of lobar fissures in CT images of the lung. A similar approach that employs local weighting and direction-dependent anisotropic diffusion smoothing resulted in more realistic displacement fields than methods using global smoothing regularization [26]. Alternatively, sliding motion in DIR has been approached by using novel transformation models that allow for discontinuities at predefined boundaries. One such example is the use of a linear combination of multiple B-spline functions and a sliding constraint [27]. This enhanced formulation of the classical free-form deformation (FFD) model [3] delivered accurate estimations of the displacement deformation field in 16 patients with lung cancer. A step further is the extension of the FFD free-form deformation method, which consists in enhancing B-spline basis functions with discontinuous functions that have jumps defined at the discontinuity surface [22,24], a formulation that has been termed extended FFD (XFFD). XFFD has shown to deliver high accuracy when registering synthetic images with strong sliding discontinuities, as well as lung and liver images where high levels of sliding are present. While incorporating additional constraints that account for organ sliding results in better

deformation estimates, several limitations preclude them from their direct applicability in the analysis of regional strain deformation in the lungs. A key limitation presented by XFFD models, which is also shared by other B-spline methods [27,29] and diffusion-based methods [25,26], is the fact that they rely on the definition of the sliding boundaries prior to the DIR analysis, which is typically done using semi-automatic segmentation methods, and largely depends on expert knowledge of the spatial location of the discontinuity boundaries. The a-priori definition of the sliding boundaries and the need for fine grids to capture curved surfaces where sliding occurs largely limit the applicability of current DIR methods in registering lung images, where the pleural cavity and fissures have intricate surface geometries and may not be easy to detect by the non-expert user.

The scientific question that motivates this work reads as follows: Is it possible to accurately capture sliding motion in DIR without the predefined knowledge of the sliding boundaries? To answer this question, in this work we aim at proposing and validating an inelastic DIR (i-DIR) method that allows for the automatic detection of sliding boundaries and that can handle discontinuous sliding motion on such surfaces.

## 2. Materials and Methods

### 2.1. Deformable Image Registration Elastic Formulation

In the following we adopt a variational framework for DIR problems [2,30], which will be the starting point of the i-DIR formulation. Let $\Omega \subset \mathbb{R}^n$ be a domain of interest (image support), $R : \Omega \to \mathbb{R}$ be the reference image and $T : \Omega \to \mathbb{R}$ be the target image. The DIR problem aims to establish an optimal transformation $u : \Omega \to \mathbb{R}^n$ that best aligns the reference and target images. To this end, we consider the functional space $\mathcal{V} := \boldsymbol{H}^1(\Omega, \mathbb{R}^n)$ and define a similarity functional $\mathcal{D} : \mathcal{V} \to \mathbb{R}$ that penalizes differences between the reference image $R$ and the resampled target image $T \circ (\boldsymbol{id} + \boldsymbol{u})$. A popular choice for the similarity measure in mono-modal applications of DIR [31,32] is the sum of squared-differences

$$\mathcal{D}[\boldsymbol{w}] := \frac{1}{2}\int_\Omega [T(\boldsymbol{x} + \boldsymbol{w}(\boldsymbol{x})) - R(\boldsymbol{x})]^2 d\Omega \quad , \forall \boldsymbol{w} \in \mathcal{V}, \tag{1}$$

which we will consider throughout this work. We remark that other choices of image similarity models such as those based on cross-correlation and mutual information measures can also be included in this formulation [30,33]. Further, we define a regularizer $\mathcal{S} : \mathcal{V} \to \mathbb{R}$ that provides smoothness to the optimal transformation as well as it avoids ill-posedness of the DIR problem. A popular choice due to its physical meaning is the elastic regularizer

$$\mathcal{S}[\boldsymbol{w}] := \int_\Omega W^e(\nabla \boldsymbol{w}) d\Omega, \tag{2}$$

where the elastic energy density takes the form

$$W^e(\nabla \boldsymbol{w}) := \mu \|\nabla \boldsymbol{w} + \nabla \boldsymbol{w}^T\|^2 + \frac{\lambda}{2}(\text{div } \boldsymbol{w})^2, \tag{3}$$

with $\nabla$ the gradient operator, div the divergence operator, and $\lambda$ and $\mu$ the Lamé constants. With these elements, we define the elastic registration functional as

$$\Pi[\boldsymbol{w}] := \alpha \mathcal{D}[\boldsymbol{w}] + \mathcal{S}[\boldsymbol{w}], \tag{4}$$

where $\alpha > 0$ is a weighting parameter. Then, the optimal transformation $\boldsymbol{u}$ is the minimizer of the elastic registration functional, and the DIR problem is formulated as the following variational problem: Find $\boldsymbol{u}$ such that

$$\Pi[\boldsymbol{u}] = \min_{\boldsymbol{w} \in \mathcal{V}} \Pi[\boldsymbol{w}]. \tag{5}$$

We note that the optimal transformation $\boldsymbol{u}$ can be interpreted as a displacement field that maps a point between its locations in the reference and target images. Moreover, the choice of the elastic deformation energy as a regularization term confers the DIR problem the physical interpretation of an elasticity problem [34], which has been widely exploited in the literature [1]. Further, and based upon this physical interpretation of the optimal transformation $\boldsymbol{u}$, we define the strain tensor operator

$$\boldsymbol{\varepsilon}(\boldsymbol{u}) := \frac{1}{2}\left(\nabla\boldsymbol{u} + \nabla\boldsymbol{u}^T\right), \tag{6}$$

and we note that the elastic energy density defined in (3) can be rewritten as

$$W^e(\boldsymbol{\varepsilon}) = \mu\boldsymbol{\varepsilon} : \boldsymbol{\varepsilon} + \frac{\lambda}{2}(\text{trace }\boldsymbol{\varepsilon})^2, \tag{7}$$

where : signifies the tensor scalar (inner) product and trace represents the trace operator. We further define the stress tensor associated to this elastic energy by

$$\sigma(\boldsymbol{\varepsilon}) := \frac{\partial W^e}{\partial \boldsymbol{\varepsilon}} = 2\mu\boldsymbol{\varepsilon} + \lambda \text{ trace }(\boldsymbol{\varepsilon})\boldsymbol{I}, \tag{8}$$

where $\boldsymbol{I}$ is the identity tensor.

### 2.2. The Inelastic Deformable Image Registration (I-Dir) Method

As discussed in the introduction, the elastic regularizer is not suited to handle discontinuities in the displacement field. As a result, sliding motion is not captured by traditional DIR methods. To address this limitation, here we draw ideas from the mechanics of inelastic solids, which aims at modeling inelastic deformation processes that result in localized softening in a solid. In the following, we briefly summarize the main ingredients of a traditional von Mises plasticity model, for a comprehensive review of the theory of inelastic solids we refer the reader to [35]. Inelastic deformation in metals is driven by shearing deformation mechanisms, where sliding in the plane of maximum shearing occurs when the shear stress in that plane overcomes a critical yield stress, resembling frictional sliding motion.

We note that the mechanical behavior of an inelastic solid is path dependent, which we represent through a time-dependence of the associated displacement field and strain tensors. To model inelastic deformation processes, we adopt the standard additive decomposition of the strain tensor

$$\boldsymbol{\varepsilon} = \boldsymbol{\varepsilon}^e + \boldsymbol{\varepsilon}^p \tag{9}$$

where $\boldsymbol{\varepsilon}^e$ corresponds to the elastic strain tensor which is assumed to disappear as the load is removed, and $\boldsymbol{\varepsilon}^p$ is the inelastic strain tensor which captures permanent deformations (i.e., sliding) that will remain in the solid after the load is removed. A sketch of this traditional decomposition of deformations is included in Figure 1. For the purposes of image registration, we note the inelastic strain tensor will capture sliding that does not generate deformation in a tissue, and therefore we will quantify regional deformation solely based on the elastic strain tensor. The additive decomposition carries onto the instantaneous evolution of strain components, and we note that (9) implies that

$$\dot{\boldsymbol{\varepsilon}} = \dot{\boldsymbol{\varepsilon}}^e + \dot{\boldsymbol{\varepsilon}}^p, \tag{10}$$

where $\dot{(\;)}$ indicates partial derivatives with respect to time. To reflect the path-dependent nature of inelastic solids, we consider the effective inelastic strain $q \in \mathcal{M} := L^2(\Omega, \mathbb{R})$ as the hardening internal variable. Following a thermodynamic formalism, we assume a free energy density function (for rate-independent plasticity) that extends the elastic energy density (7) and takes the form

$$A(\varepsilon, \varepsilon^p, q) = W^e(\varepsilon - \varepsilon^p) + W^p(q), \tag{11}$$

where we assume that the stored plastic energy takes the form $W^p = \frac{1}{2}Hq^2$, with $H$ being the hardening modulus. Then, the elastic constitutive relation reads

$$\sigma(\varepsilon^e) = \frac{\partial W}{\partial \varepsilon^e} = 2\mu\varepsilon^e + \lambda \, \text{trace}\,(\varepsilon^e)I, \tag{12}$$

and the relation between the hardening internal variable and its thermodynamic conjugate stress is

$$\sigma_c(q) = \frac{\partial A}{\partial q} = Hq. \tag{13}$$

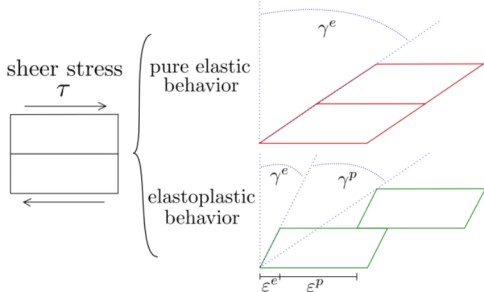

**Figure 1.** Schematics comparing the behaviour between an elastic and an inelastic approach.

The inelastic behavior of a solid is modeled by defining a inelastic potential, also known as the yield function, which in the case of a von Mises solid takes the form

$$\Phi(\sigma, \sigma_c) := \sqrt{\frac{3}{2} s_{ij} s_{ij}} - \sigma_c \tag{14}$$

where $s_{ij}$ are the components of the deviatoric stress tensor $s$ defined as

$$s(\sigma) := \sigma - \frac{1}{2} \, \text{trace}\, \sigma I. \tag{15}$$

Following an associative plasticity framework [35], the evolution of the inelastic strain tensor is governed by the flow rule

$$\dot{\varepsilon}^p = \dot{\gamma} \frac{\partial \Phi}{\partial \sigma}(\sigma, \sigma_c) = \dot{\gamma} \sqrt{\frac{3}{2}} \frac{s}{\|s\|} \tag{16}$$

where $\dot{\gamma}$ is the inelastic multiplier and the norm defined as $\|(\cdot)\| = \sqrt{(\cdot) : (\cdot)}$. The evolution of internal variables is also dictated by the gradients of the inelastic potential

$$\dot{q} = -\dot{\gamma} \frac{\partial \Phi}{\partial \sigma_c}(\sigma, \sigma_c) = \dot{\gamma}. \tag{17}$$

Finally, an inelastic model must comply with the loading/unloading complimentary conditions

$$\Phi(\sigma, \sigma_c) \leq 0, \qquad \dot{\gamma} \geq 0, \qquad \dot{\gamma}\Phi(\sigma, \sigma_c) = 0. \tag{18}$$

Complimentary conditions (18) are interpreted as follows: Inelastic deformations will occur ($\dot{\gamma} > 0$) only when the current stress state reaches the yield surface $\Phi(\sigma, \sigma_c) = 0$. Otherwise, the inelastic evolution must be null to comply with (18), i.e., $\dot{\gamma} = 0$, which in turn implies that the change in total deformation will correspond only to changes in elastic deformation, as governed by (10).

In general, the relationship between $\dot{\varepsilon}^p$ and $q$ is not independent. Let $\bar{\sigma}$ be the effective stress defined as

$$\bar{\sigma} = \sqrt{\frac{3}{2}s_{ij}s_{ij}}. \tag{19}$$

Then, the relation between $\dot{\varepsilon}^p$ and $q$ is assumed to follow the Prandtl-Reuss flow rule that reads

$$\dot{\varepsilon}^p_{ij} = \dot{\bar{\varepsilon}}^p\left(\frac{3}{2}\frac{s_{ij}}{\bar{\sigma}}\right), \tag{20}$$

where $\dot{\bar{\varepsilon}}^p$ stands for the evolution of the accumulated plastic strain. In view of the plastic flow rule, the accumulated plastic strain is equivalent to $q$, so by integration of (20) and assuming isotropic hardening, we have the following relation,

$$\bar{\varepsilon}^p = \int_0^t \sqrt{\frac{2}{3}\dot{\varepsilon}^p_{ij}\dot{\varepsilon}^p_{ij}}\, dt \equiv q \tag{21}$$

(see Appendix A.1 for details).

### 2.3. Time and Space Discretization

Given $\dot{\varepsilon}$, the set of Equations (10), (12), (13), (16) and (18) constitutes an inelastic constitutive initial value problem, which has been traditionally solved using a return-mapping algorithm based on an implicit backward-Euler temporal discretization. To this end, the time variable is discretized in generic subintervals $[t_n, t_{n+1}]$. Then, a series of incremental problems are obtained, where the main variables of the inelasticity model are assumed to be known at the time $t = t_n$, and need to be solved for $t = t_{n+1}$, giving rise to classical return-mapping algorithms. The details about the numerical discretization of return mapping algorithms can be found elsewhere [36]. Conveniently, the elastoplastic incremental problems can be reformulated as incremental variational (minimization) problems, which gives rise to the theory of variational updates in the computational solid mechanics community [37–39]. In the following, we draw ideas from the theory of variational updates in plasticity to formulate the inelastic DIR model. The general framework consists in formulating the evolution of an elastoplastic solid as a sequence of incremental variational minimization problems. To this end, we first integrate the flow rule (16) using a Backward-Euler scheme to obtain

$$\varepsilon^p_{n+1} - \varepsilon^p_n = (q_{n+1} - q_n)\sqrt{\frac{3}{2}}\frac{s(\varepsilon_{n+1} - \varepsilon^p_{n+1})}{\|s(\varepsilon_{n+1} - \varepsilon^p_{n+1})\|} \tag{22}$$

Solving for $\varepsilon^p_{n+1}$ from the non-linear Equation (22) delivers an incremental update for the inelastic strain tensor, which we express as

$$\varepsilon^p_{n+1} = \varepsilon^{*p}_{n+1}(\varepsilon_{n+1}, q_{n+1}) \tag{23}$$

which depends solely of $\varepsilon_{n+1}$ and $q_{n+1}$. Based on this flow-rule update, we define the effective incremental energy density for $t = t_n$ as [38],

$$W_n(\varepsilon) = \inf_{q_{n+1}} g_n(\varepsilon, q_{n+1}) \tag{24}$$

where,

$$g_n(\varepsilon, q_{n+1}) = A(\varepsilon, \varepsilon_{n+1}^{*p}(\varepsilon, q_{n+1}), q_{n+1}) - A_n + \Delta t \cdot \psi^* \left( \frac{|q_{n+1} - q_n|}{\Delta t} \right) \tag{25}$$

where $\Delta t = t_{n+1} - t_n$, and $\psi^*$ stands for the dual dissipation potential [38] that governs the time evolution of the hardening variable, which in our case is defined as $\psi^* = \sigma_y |\Delta q|$, with $\Delta q = q_{n+1} - q_n$. The minimization problem involved in the definition (25) is equivalent to the stationary condition

$$0 \in \frac{\partial A}{\partial q_{n+1}} + \partial \psi^* \left( \frac{|q_{n+1} - q_n|}{\Delta t} \right) \tag{26}$$

which, for the rate-independent case reads (see details in Appendix A.2),

$$\bar{\sigma}_{n+1}^{pre} - 3\mu \Delta q - \sigma_c(q_{n+1}) = \partial \psi^* \left( \frac{|q_{n+1} - q_n|}{\Delta t} \right) \tag{27}$$

where $\bar{\sigma}_{n+1}^{pre}$ is the elastic predictor for $\bar{\sigma}_{n+1}$. Substituting (13) into (27) we obtain

$$\bar{\sigma}_{n+1}^{pre} - 3\mu \Delta q - H(q_n + \Delta q) - \partial \psi^* \frac{|\Delta q|}{\Delta t} = 0 \tag{28}$$

where the sub-differential of $\psi^*$ is defined as:

$$\partial \psi^* = \begin{cases} [-\sigma_y, \sigma_y] & \text{if} \quad \Delta q = 0 \\ \sigma_y & \text{if} \quad \Delta q > 0 \\ -\sigma_y & \text{if} \quad \Delta q < 0 \end{cases} \tag{29}$$

The solution of $\Delta q$ from (28) involves two mutually exclusive steps, giving rise to a return-mapping algorithm which involves an elastic predictor and a plastic corrector steps, see Appendix A.3.

With the definition of the effective incremental energy density, we now postulate the inelastic DIR formulation as a sequence of effective variational problems. For a generic time step, the displacement field $u_n$ is assumed to be known, and we find the displacement field $u_{n+1}$ by solving the problem

$$\Pi_n^{\text{eff}}[u_{n+1}] = \min_{w \in \mathcal{V}} \Pi_n^{\text{eff}}[w], \tag{30}$$

where the inelastic DIR functional reads

$$\Pi_n^{\text{eff}}[w] = \mathcal{S}_n^{\text{eff}}[w] + \alpha \mathcal{D}[w], \tag{31}$$

and the inelastic regularizing term takes the form

$$\mathcal{S}_n^{\text{eff}}[w] = \int_\Omega W_n(\varepsilon(w)). \tag{32}$$

To solve the minimization problem (30) we consider the stationary condition

$$\mathbf{R}_n[w; v] := \frac{d}{d\epsilon} \Pi_n^{\text{eff}}[w + \epsilon v]_{\epsilon=0} = 0, \qquad \forall v \in \mathcal{V}. \tag{33}$$

The residual in (33) takes the form

$$\mathbf{R}_n[\boldsymbol{w};\boldsymbol{v}] := \alpha \int_\Omega \boldsymbol{v} \cdot (T(\boldsymbol{w}) - R)\nabla T(\boldsymbol{w}) + \int_\Omega \varepsilon(\boldsymbol{v}) : \sigma_{n+1}(\varepsilon(\boldsymbol{w})), \qquad (34)$$

where

$$\sigma_{n+1}(\boldsymbol{\varepsilon}) := \frac{\partial W_n}{\partial \boldsymbol{\varepsilon}}(\boldsymbol{\varepsilon}), \qquad (35)$$

represents the stress tensor update [38]. The residual Equation (33) constitutes a nonlinear problem, which we approach by means of linearization. To this end, we consider the Gauteaux differential defined as

$$\mathbf{TR}[\boldsymbol{w},\boldsymbol{v};\Delta\boldsymbol{w}] := \alpha \int_\Omega \boldsymbol{v} \cdot \left\{ \nabla T(\boldsymbol{w}) \otimes \nabla T(\boldsymbol{w}) + (T(\boldsymbol{w}) - R)\nabla\nabla T(\boldsymbol{w}) \right\} \cdot \Delta\boldsymbol{w} + \int_\Omega \varepsilon(\boldsymbol{v}) : \mathbf{D}^{ep}_{n+1}\,\varepsilon(\Delta\boldsymbol{w}), \qquad (36)$$

where

$$\mathbf{D}^{ep}_{n+1}(\boldsymbol{\varepsilon}) := \frac{\partial^2 W_n}{\partial \boldsymbol{\varepsilon}^2}, \qquad (37)$$

is the consistent tangent tensor, see Appendix A.4. Thus, the linearized version of the residual problem reads: Given an initial guess $\boldsymbol{w} \in \mathcal{V}$, find the increment $\Delta\boldsymbol{w}$ such that

$$\mathbf{R}_n[\boldsymbol{w};\boldsymbol{v}] + \mathbf{TR}[\boldsymbol{w},\boldsymbol{v};\Delta\boldsymbol{w}] = 0 \quad \forall \quad \boldsymbol{v} \in \mathcal{V}, \qquad (38)$$

and we iterate over this linearized problem until a convergence criterion is reached.

To solve the continuous linear variational problem defined in (38) we adopt a Ritz-Galerkin finite-element approach. To this end, we construct the finite-element space

$$\mathcal{V}^h = \left\{ v^h : \Omega^h \to \mathbb{R}^n \,|\, v^h := \sum_{A=1}^m N_A v_A, \text{with } v_A \in \mathbb{R}^n \right\} \subset \mathcal{V}, \qquad (39)$$

where $\{N_1, \ldots, N_m\}$ is the set of basis functions. Using this finite-element space, we approximately solve the variational problem (38), i.e., we solve the problem: Given an initial guess $\boldsymbol{u}^h$, find the increment $\Delta\boldsymbol{u}^h$ such that

$$\mathbf{R}_n[\boldsymbol{w}^h;\boldsymbol{v}^h] + \mathbf{TR}_n[\boldsymbol{u}^h,\boldsymbol{v}^h;\Delta\boldsymbol{u}^h] = 0 \quad \forall \quad \boldsymbol{v}^h \in \mathcal{V}^h. \qquad (40)$$

Using standard arguments (e.g., see [40]) we can show that (40) is equivalent to solving the linear system of equations

$$\mathbf{K}_n \Delta\mathbf{u} = \mathbf{F}_n, \qquad (41)$$

where $\Delta\mathbf{u}$ is a vector with the nodal values of the increment $\Delta\boldsymbol{u}^h$, $\mathbf{K}_n$ is the tangent matrix and $\mathbf{R}_n$ is the residual for the previous guess, all of which are defined in Appendix A.5. After convergence is reached for the Newton step, the internal variables $q$ at $t_{n+1}$ are updated and stored at the element level. We further note that, in order to provide stability and unisolvence of the problem, we adopt the approach set forth in [30], where we impose orthogonality conditions to the displacement fields and assume Neumann boundary conditions.

### 2.4. Performance Assessment and Metrics

The i-DIR method was implemented using an in-house Python code. In order to contrast the results of the i-DIR method with other DIR methods, we considered the open source Nifty Reg library [41] which efficiently implements the FFD method [3] with elastic regularization. To understand the effect of the inelastic regularization term over the purely elastic counterpart for a FE method, we also consider the comparison with an Elastic FEM registration. To study the performance of the three methods considered here (FFD, Elastic FEM, and i-DIR), we constructed synthetic reference and target images that simulated planar sliding over a chessboard-like image studied by Rua and co-workers [22], which we refer to as the synthetic dataset with sliding motion, see Figure 2. The synthetic images

have a resolution of $80 \times 80$ pixels, and the target image is constructed in such a way that it resembled a dislocation or sliding motion, with a known displacement of 5 pixels. We remark that, as the motion corresponds to a uniform vertical displacement (i.e., rigid body motion in both blocks) of the right-hand side of the image, the exact strain field is equal to zero, as rigid motions do not generate strain. To assess the method's performance on anatomical images, we considered sagittal planes of CT thorax images of a normal volunteer under spontaneous breathing at total lung capacity (reference image) and functional residual capacity (target image), see Figure 3. The images were randomly selected from a small CT lung dataset of normal subjects employed in a previous study [13], and the sagittal planes were arbitrarily chosen so that large deformations were explicitly depicted to capture sliding.

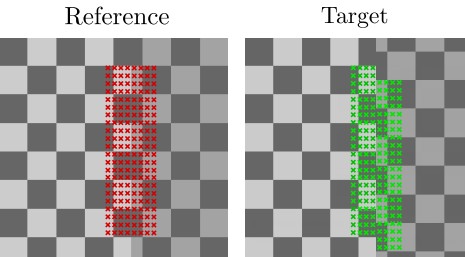

**Figure 2.** Synthetic dataset with sliding motion: Reference (**left**) and target (**right**) images. Landmarks used for computing the target registration error (TRE) are shown in red for the reference image and in green for the target image.

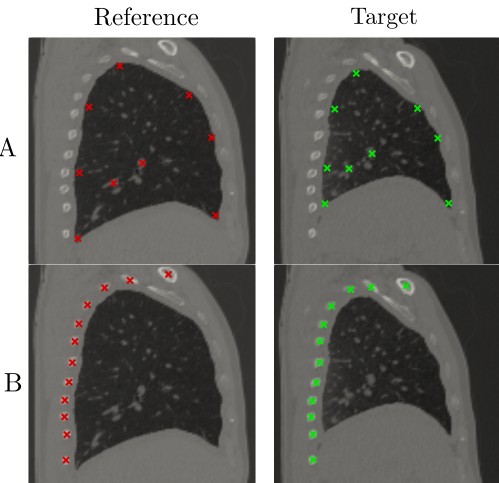

**Figure 3.** Lung dataset: Reference (**left**) and target (**right**) images. (Row **A**): TRE analysis using landmarks inside the lung. (Row **B**): TRE analysis using using landmarks on the dorsal ribs. Red and green marks indicate landmarks in the reference and target images, respectively.

To quantitatively evaluate the performance of the DIR methods, we considered the traditional residual sum of squared differences (RSS) between the reference and resampled images, defined as:

$$RSS = \sum_{i=1}^{m} \sum_{j=1}^{n} (R_{ij} - (T \circ (\boldsymbol{id} + \boldsymbol{u}))_{ij})^2. \tag{42}$$

In addition, the normalized target registration error (TRE) was also computed. The TRE is defined as

$$TRE = \frac{\sum_{i=1}^{N} \sqrt{(p_i - q_i)^2}}{N} \tag{43}$$

where $p_i(x, y)$ and $q_i(x, y)$ are the $i$th landmark in the target image (fixed landmark) and the moving landmark, respectively, and $N$ is the total number of landmarks. For the synthetic

dataset, 375 landmarks were positioned around the discontinuity surface, as shown in Figure 2. In the case of the lung dataset, two sets of landmarks were considered, to analyze the effect of landmark selection and position regarding the sliding surface. The first case considered 9 landmarks positioned entirely inside the lung (Figure 3, top row). The second case considered 12 landmarks placed in the ribs of the dorsal region (Figure 3, bottom row).

In addition to the RSS and TRE metrics, the resampled image ($T \circ (id + u)$), difference image ($R - T \circ (id + u)$), and warped reference image ($(\varphi, R)$) are reported for all cases. To study the mechanical performance of all the methods studied, we constructed images of the elastic volumetric strain, defined as

$$\varepsilon_{vol}^{e} := \text{trace}(\boldsymbol{\varepsilon}^{e}), \tag{44}$$

and images of the elastic von Mises strain, which takes the form

$$\varepsilon_{vm}^{e} := \sqrt{\frac{2}{3}\boldsymbol{\varepsilon}^{e} : \boldsymbol{\varepsilon}^{e}}. \tag{45}$$

We note that for purely elastic methods (FFD, Elastic FEM), the elastic strain tensor $\boldsymbol{\varepsilon}^{e}$ is replaced by the total strain tensor $\boldsymbol{\varepsilon}$ in (44) and (45). Finally, a sensitivity analysis is conducted for the i-DIR method on the synthetic dataset to understand the effect of the initial yield stress on its registration performance.

### 2.5. Parameter Settings

For the FEM models (elastic FEM and i-DIR), we established an incremental approach for the weighting parameter $\alpha$. We set an initial value of $\alpha = 0.01$, and once a convergence tolerance was exceeded, we systematically increased $\alpha$, until we reach a value of $\alpha = 1400$. Values for the Lamé constants ($\mu, \lambda$), the initial yield limit ($\sigma_0$) and the hardening modulus ($H$) are included in Table 1.

**Table 1.** Parameter values for the inelastic regularizer.

| Parameter | Value |
|:---:|:---:|
| $\mu$ | 1.36 |
| $\lambda$ | 0.34 |
| $\sigma_0$ | 0.1 |
| $H$ | 0.3 |

Following the pyramidal approach described in [41], the FFD model consisted of a global registration and three consecutive local registration processes. The penalty term associated with the FFD model, also known as bending energy (BE), was set to: $BE_1 = 1 \times 10^{-9}$, $BE_2 = 2.5 \times 10^{-6}$ and $BE_3 = 1 \times 10^{-4}$, for the three local registration respectively.

In terms of numerical discretization, both the elastic FEM and i-DIR models, employed structured triangular finite element meshes. As shown in Figure 4 the synthetic dataset used a mesh of size 5618 elements and for the lung dataset a mesh of size 18,860 elements. For visualization purposes, we further refined our results into structured meshes of size 64,800 and 28,800 elements for the synthetic and lung dataset, respectively. As for the FFD model, we projected the deformation mapping field (output) and compute the mechanical measures into a refined structured triangular finite element mesh of the same size as the FEM DIR models, in order to have a fair comparison.

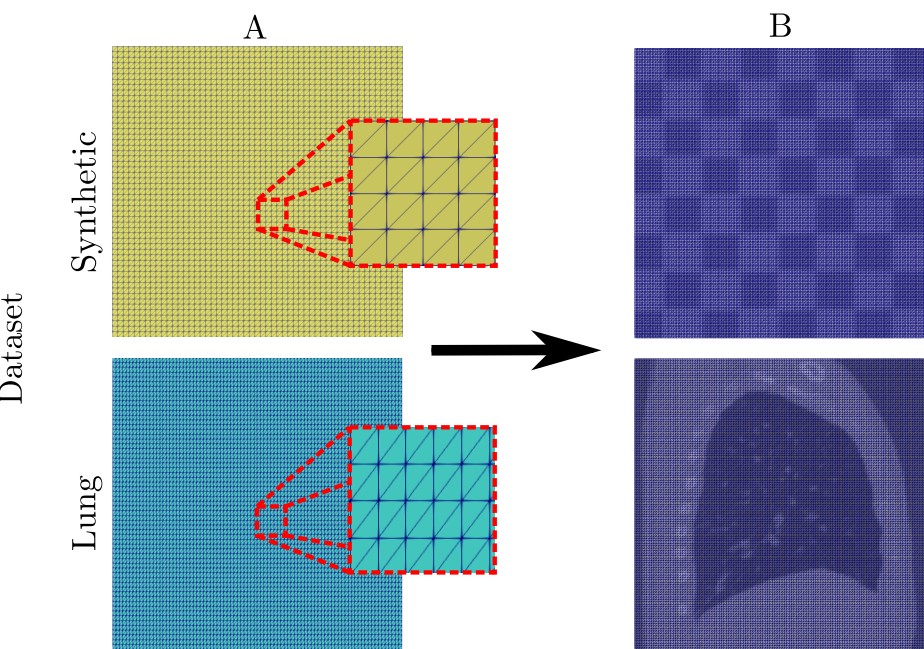

**Figure 4.** Numerical discretization of the synthetic and lung datasets. (Column **A**) Structured triangular meshes and (Column **B**) refined structured triangular meshes.

### 3. Results

*3.1. Synthetic Dataset with Sliding Motion*

The performance of each registration model using the synthetic dataset with sliding motion is reported in terms of resampled and difference images in Figure 5. The i-DIR method accurately captures the vertical sliding and delivers the best resampled image, when compared to the other elastic methods (Figure 5, top row). Most of the errors in the resampled images are located in a small neighborhood of the line where sliding takes place. When analyzing the difference images (Figure 5, bottom row), the FFD method results in considerable voxel-wise differences at the boundaries of the squares that propagate from the sliding line throughout the checkerboard domain. In contrast, small differences are observed around the sliding line in the elastic FEM case. No visible differences are observed for the i-DIR case when compared to the other two methods.

Warped reference images showing the resulting displacement field for each method are reported in Figure 6. A close-up around the sliding region shows a continuous displacement field with a vortex-like pattern over the sliding line that slowly dissipates to the right for the FFD case. A similar displacement field pattern is observed for the elastic FEM case, but with an attenuated vortex pattern. In contrast, the i-DIR method delivers a uniformly vertical displacement field on the region to the right of the sliding line, and zero displacements to the left of the sliding line, being able to identify the discontinuity surface as well as capturing the discontinuous displacement field.

The RSS and TRE metrics for all three methods are shown in Table 2. The i-DIR method delivers the lowest values for these performance metrics, followed by the Elastic FEM method.

**Table 2.** Performance metrics for the synthetic dataset.

| Model | RSS | TRE |
|---|---|---|
| FFD | 16.32 | 1.17 |
| Elastic FEM | 3.19 | 0.46 |
| i-DIR | 0.28 | 0.22 |

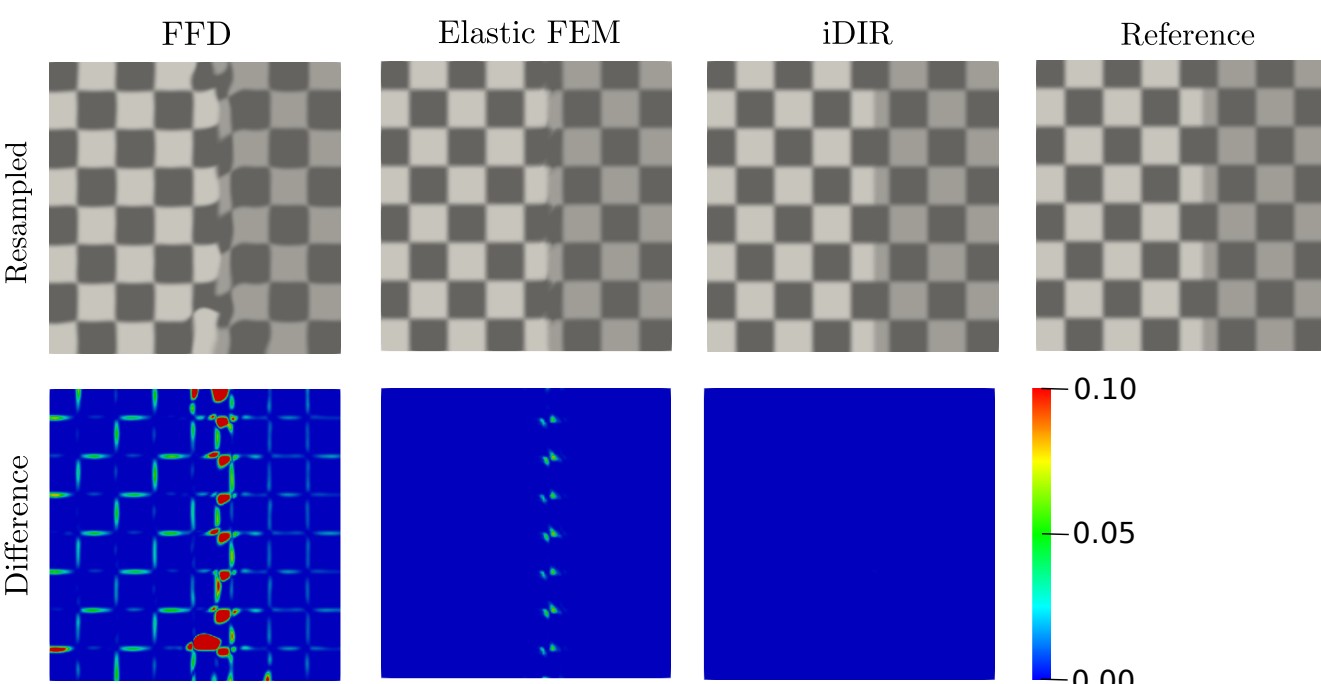

**Figure 5.** Registration of synthetic dataset with sliding motion. (**Top row**) resampled images using free-form deformation (FFD), Elastic finite element method and inelastic deformable image registration (iDIR) methods and reference image, (**bottom row**) difference images. Colorbar indicates the absolute intensity difference between images.

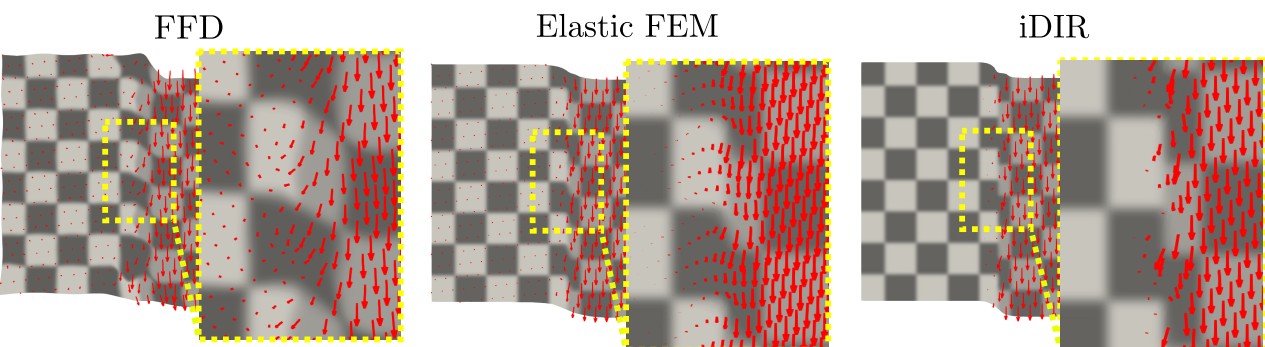

**Figure 6.** Warped reference image and displacement field for the synthetic dataset with sliding motion. Red arrows show the displacement field in a neighborhood of the sliding plane.

Figure 7 shows the elastic deformation fields associated to the three registration methods. The elastic volumetric strain displayed by the FFD model, shows an erratic pattern throughout the entire image, with high values of both compressive (peak value of $-1.22$) and expansive (peak value of $0.83$) deformation near the vicinity of the discontinuity surface. The case of the elastic FEM model delivers a volumetric strain field with localized strain concentrations around the sliding surface with peak values of $-0.75$ and $1.27$. In contrast, the i-DIR model delivers a volumetric strain field that is zero in the majority of the region of analysis, with small concentrations around the sliding surface with peak values of $-0.39$ and $0.63$. The resulting elastic von Mises strain field, which characterizes shear distortions, is shown in the bottom row of Figure 7. Similarly to the case of volumetric strain, the FFD method results in a highly oscillating field that take on non-zero values everywhere in the image domain, reaching peak values of $1.72$. The Elastic FEM method displays high strain concentrations around the sliding plane with peak von Mises strain values that are similar to the FFD case ($2.38$), but the strain field rapidly dissipate away from the discontinuity plane. In contrast, the i-DIR results in a narrow region around

the sliding plane with low values (peak value of 0.38), with the rest of the image domain resulting in zero von Mises strain.

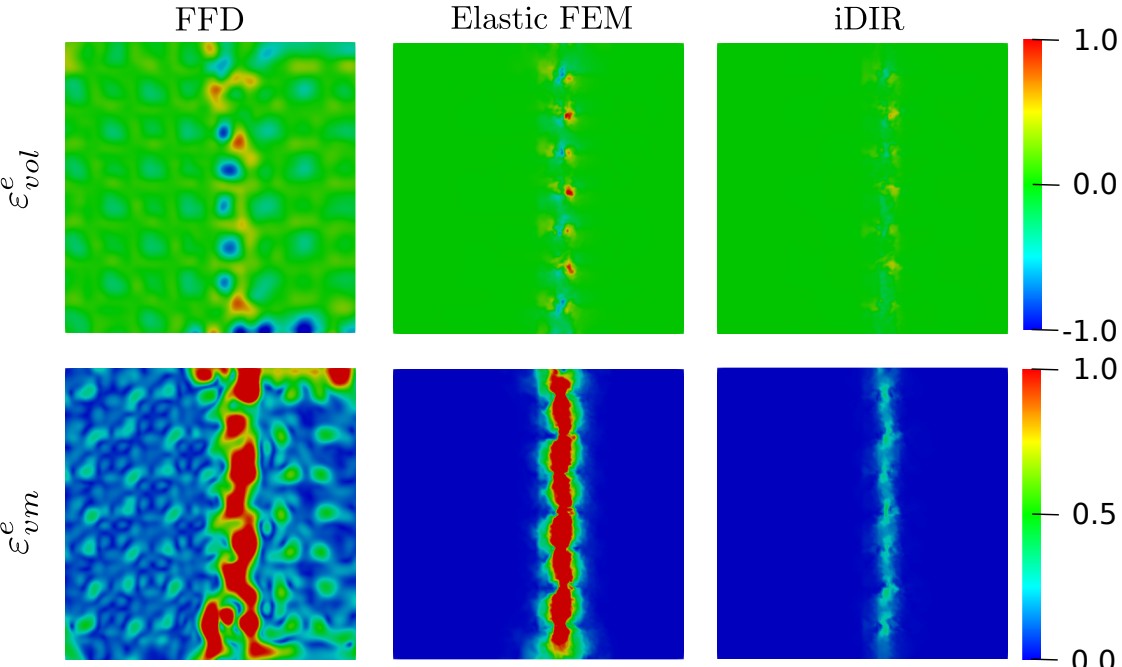

**Figure 7.** Elastic deformation fields for the synthetic dataset with sliding motion resulting from the different registration methods: elastic volumetric strain (**top row**, colorbar displays strain magnitude), and elastic von Mises strain (**bottom row**, colorbar displays strain magnitude).

The sensitivity of the RSS error to the value of the initial yield stress in the i-DIR method is shown in Figure 8. Yield stress values smaller that 0.1 result in RSS errors that do no change considerably, delivering the highest accuracy observed for all three methods. In contrast, yield stress values above 1.0 deliver a much higher RSS error, which approaches that of the Elastic FEM method, see Table 2.

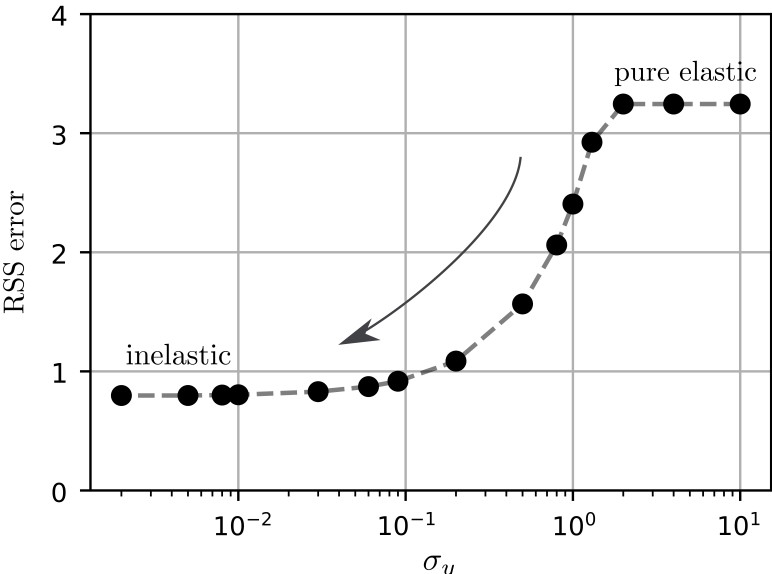

**Figure 8.** Sensitivity of the i-DIR method, measured in terms of residual sum of squared differences (RSS) error, to the choice of initial yield stress value.

*3.2. Registration of Lung CT Images*

Resampled and difference images for the lung CT dataset are shown in Figure 9, top row. All three methods deliver similar results of the resampled image. We note however, that resampled images from both the FFD and Elastic FE methods show distorted rib cuts in the dorsal region, while the ribs are accurately resampled in the case of the i-DIR method. The misalignment of the ribs is also observed in the difference images of the FFD and Elastic FEM cases, see Figure 9, bottom row. In contrast, the i-DIR case reports zero difference values in the regions where ribs are located.

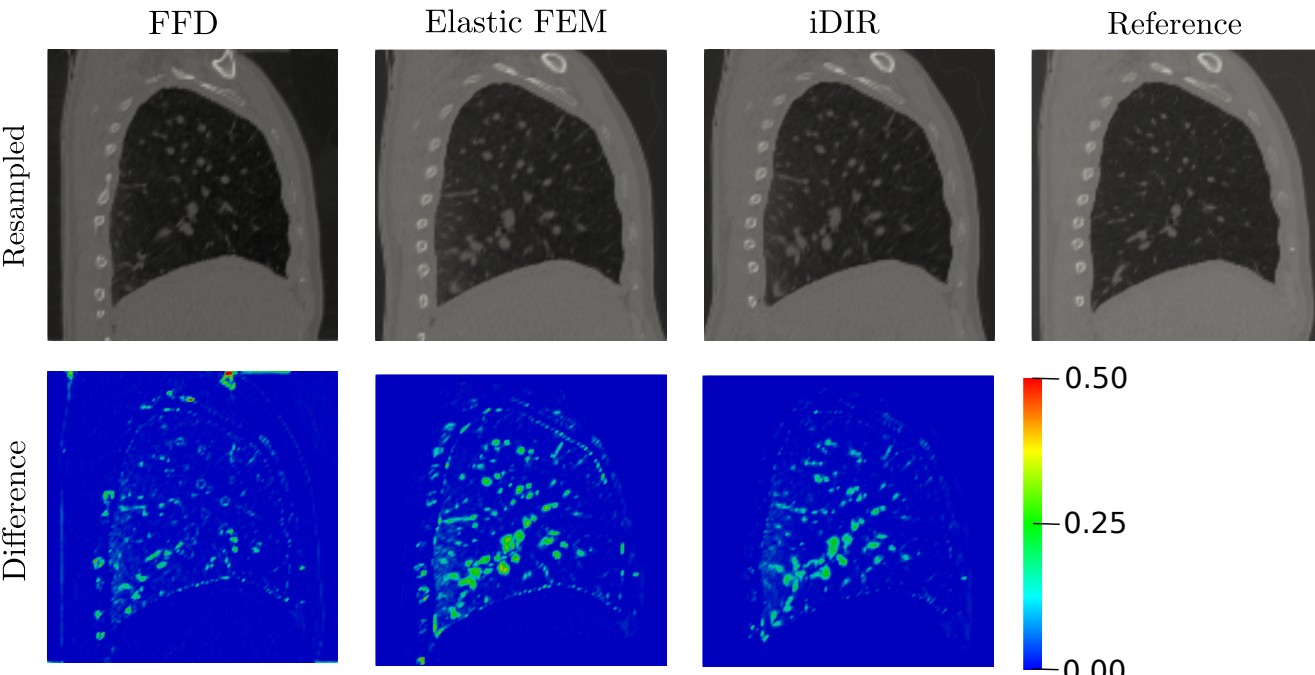

**Figure 9.** Registration of the lung dataset and comparison between methods. (**Top row**) resampled images, (**bottom row**) difference images. Colorbar indicates the absolute intensity difference between images. Reference image is included for comparison purposes.

Warped reference images are shown in Figure 10, where a close-up shows the displacement fields around the sliding pleural cavity. A continuous, and almost uniform upward displacement field is observed for the case of the FFD and Elastic FE methods. In contrast, the i-DIR method delivers an upward displacement field inside the lung, right next to a region comprising the ribs with null displacement, with the jump in displacement magnitude located on the sliding pleural cavity.

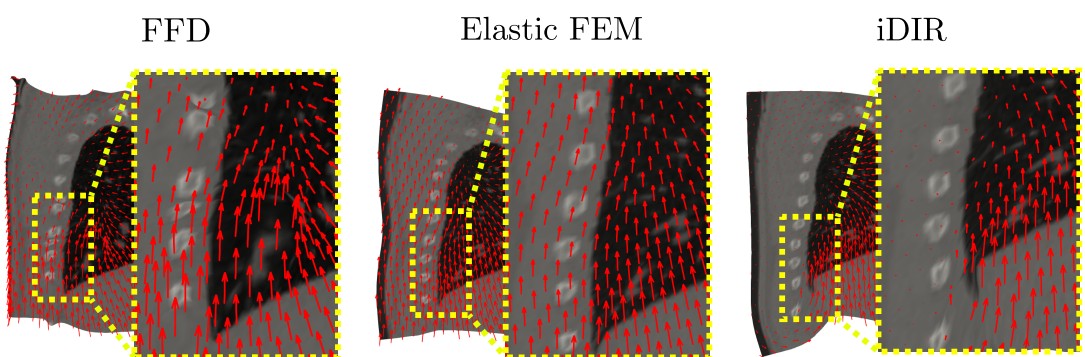

**Figure 10.** Warped reference image and displacement field for the lung dataset. Red arrows show the displacement field in a neighborhood of the sliding surface.

Performance metrics for the lung dataset are included in Table 3. We note that all three methods result in similar values for the case of RSS and TRE using inside-lung landmarks. However, the i-DIR method shows a remarkable advantage over the other methods for the case of TRE using rib landmarks.

**Table 3.** Performance metrics for the lung dataset.

| Model | RSS | TRE (Inside-Lung Landmarks) | TRE (Rib Landmarks) |
|---|---|---|---|
| FFD | 11.64 | 6.82 | 13.98 |
| Elastic FEM | 13.25 | 6.74 | 13.68 |
| i-DIR | 12.24 | 6.99 | 0.77 |

The elastic volumetric strain distribution resulting from the registration of lung images are shown in Figure 11, top row. The FFD model delivers a highly oscillating field that results in excessive strain values with peaks as high as −1.68 and 1.02, located both inside and outside the lung domain. In contrast, the Elastic FEM model displays a more uniform volumetric strain distribution inside the lung, with a smooth pattern of strain. However, high strain localizations are observed outside the lung in the dorsal region where the ribs are located, with oscillating values. The i-DIR model delivers a smooth distribution of volumetric strain inside the lung, that quickly transitions to small levels os strain immediatly outside the lung. Further, the largest strain levels are found in the regions near the diaphragm. Outside the lung, we mostly observe zero volumetric deformation throughout the remaining image domain. The von Mises strain fields are shown in Figure 11, bottom row. Similar to the case of volumetric strain, the FFD model delivers a highly oscillating field with a peak value in the order of 1.7 both outside and inside the lung. The Elastic FEM method results in a distribution with smaller strain magnitudes, which in some parts of the lung boundary are rapidly reduced to zero. In the case of the i-DIR method, a smooth distribution of non-zero strain is observed inside the lung with the highest values close to the diaphragm and dorsal region. The von Mises strain distribution sharply decays to zero in the regions outside the lung.

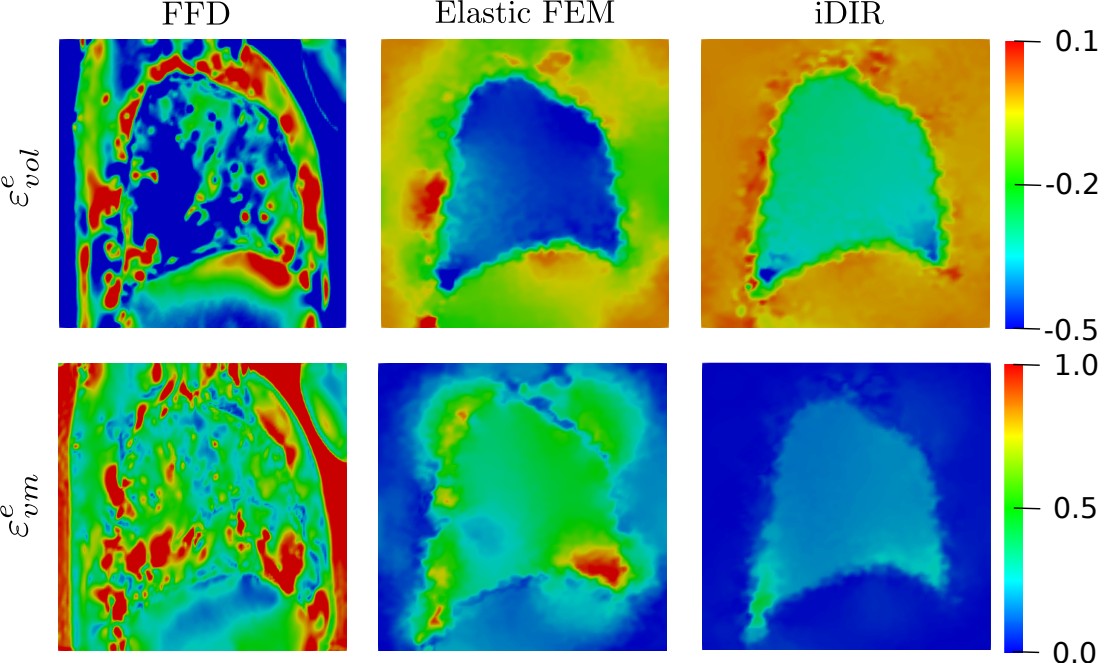

**Figure 11.** Elastic deformation fields for the lung dataset resulting from the different registration methods: elastic volumetric strain (**top row**, colorbar displays strain magnitude), and elastic von Mises strain (**bottom row**, colorbar displays strain magnitude).

## 4. Discussion

The results for the synthetic dataset with sliding motion show that among the three DIR methods studied, the i-DIR delivers the best resampled image, accurately accomodating the sliding motion, see Figure 5, top row. We note that the elastic DIR methods suffer from spurious displacements around the sliding line that result in distorted resampled images, see Figure 5, bottom row. Further, the i-DIR method is capable of capturing the discontinuous displacement field imposed by the sliding motion, while elastic DIR methods fail to capture the jump in displacements and result in spurious displacement fields, see Figure 6. Further, we have shown that for this example, the i-DIR method consistently delivers RSS and TRE metrics that confirm the superior performance of the i-DIR method when compared to the Elastic FEM and FFD methods, see Table 2. Other methods proposed in the literature have also shown a remarkable performance in the registration of the synthetic dataset with sliding motion. In particular, the XFFD method has shown to be capable of accurately capture the sliding motion by introducing a discontinuous transformation model that delivers an optimal resampling [22]. To this end, the XFFD method necessitates the definition of the sliding surface a priori in order to deliver accurate and efficient results. Here, we have shown that the i-DIR method does not require a priori information about the sliding surface. Further, the sliding plane did not coincide with any element edges in the discretization. This feature represents an important advantage over existing methods, as the i-DIR is capable of detecting sliding discontinuities in an automatic way, lending itself to the registration of images with arbitrary sliding discontinuities.

From the perspective of quantifying local deformation by means of DIR, we remark that the sliding mechanism present in the synthetic dataset corresponds to a rigid (sliding) motion between two adjacent blocks, and therefore no deformation is expected to occur in any of the blocks after sliding. Figure 7 shows that the FFD method induces spuriously high levels of both volumetric and deviatoric deformations around the sliding plane, which is consistent with previous findings for the synthetic dataset with sliding motion [22]. Similarly to the case of the displacement field, the strong warping and deformation propagates throughout the image domain, creating nonphysical high volumetric and deviatoric strain levels away from the discontinuity. The error in the strain predictions is strongly attenuated by the elastic FE method, which still concentrates high values in a neighborhood of the sliding plane. In contrast, the i-DIR method reports low levels of deviatoric deformation on a narrow band around the discontinuity surface, and negligible errors in the estimation of volumetric strain, see Figure 7. This result shows that the i-DIR method not only automatically captures the sliding motion with accuracy, but also delivers precise estimates of the strain fields, even in the present of strong discontinuities.

The sensitivity of the iDIR model to the yield stress parameter shows that for parameter values $\sigma_y \leq 10^{-2}$ no appreciable improvement is obtained in terms of the RSS error. Further, we note that high values of yield stress deliver errors that are equal to those reported by the Elastic FEM method. These results show that tuning the yield stress parameter is essential for obtaining accurate results from the registration process.

The i-DIR method was also assessed in the analysis of medical CT images of the lung, where sliding is expected to occur when registering images from resting states to maximal inspiration effort [23]. When comparing resampled images, we showed that the i-DIR method delivered errors in registering the domains inside the lung that are comparable to those found in elastic DIR methods, see Figure 9. This conclusion is supported by the performance metrics RSS and TRE for the case of inside-lung landmarks reported in Table 3, where no marked differences are observed among the FFD, Elastic FEM and i-DIR methods. However, when assessing anatomical structures that are outside the lung, i.e., ribs, we observe that the i-DIR accurately resamples them to the correct location, whereas elastic DIR methods fail to achieve a reasonable result, see Figure 9, bottom row. This observation is confirmed by the results obtained in the TRE when using rib landmarks, where the i-DIR delivers errors that are one order of magnitude smaller than the error provided by elastic DIR methods, see Table 3. Once again, we attribute the good performance of the

i-DIR method to its ability to handle discontinuous sliding motion by considering inelastic deformations in those regions, a feature not displayed by elastic methods, see Figure 10.

The evaluation of the elastic strain fields from registering lung CT images results in conclusions similar to those obtained in the case of the synthetic image: elastic DIR methods introduce highly oscillating fields both for the volumetric and deviatoric components of the elastic strain tensor, see Figure 11. We remark here that previous works on lung image registration have reported oscillatory strain fields when using elastic DIR methods that employ B-splines or other smooth and continuous basis functions for the construction of the deformation model [22,42]. However, these oscillations have been shown to hinder the accuracy of the estimations of local pulmonary deformation, as they arise due to the inability of the deformation model to capture discontinuities in the displacement field [11]. Notably, the i-DIR method delivers a smooth distribution of elastic strain inside the lung domain, with a sharp decay outside that approaches a state of no deformation. Further, the spatial patterns of volumetric strain delivered by the i-DIR method are in good agreement with those reported in the literature for normal human lungs that have been analyzed by isolating the lung domain [13], where larger volumetric strains are observed in the dorsal (dependent) and basal regions of the lung.

In conclusion, we have introduced a novel inelastic model for DIR that automatically captures sliding without a priori knowledge or assumptions about the spatial location of discontinuities or the need of a segmentation to denote the slipping domain. We note that other DIR formulations have been proposed in the literature to handle motion discontinuities without the need of segmenting the sliding region before the analysis [25]. In order to do so, these methods consider some assumptions related to the specific physiological behavior of the organ. For the case of the lungs, slipping motion is restricted to the edges of the image, and slippage occurs along the edge of the image. We note that these assumptions are not required by the i-DIR method, and therefore it represents a truly automatic technique for the detection of discontinuities in DIR of arbitrary images. The key ingredient to achieve this performance is the introduction of an inelastic energy term, which automatically locates regions of high shearing deformation associated to sliding and locally modifies the effective mechanical properties, allowing for higher levels of shear deformation in localized domains. We remark that, while inelastic formulations are standard in the field of computational mechanics [38], the inclusion of inelastic energy regularizers is novel in the field of image analysis, and, to the best of our knowledge, has not been pursued in the past in in the field of DIR. For the application of lung images, it is worth mentioning that we aim at automatically capture sliding, particularly between the lung boundary and the ribs (clearly identified within the images), and not necessarily improve the registration accuracy inside the lung. The above is supported by the results obtained when measuring the RSS error, which demonstrates that our i-DIR model holds a comparable performance with traditional DIR methods, especially in areas with no sliding. However, the inelastic model is considerably superior in capturing slippage at the lung edges, which is again substantiated by a better performance when measuring the TRE using rib-landmarks.

The present work can be extended in several directions. One limitation of the current computer implementation of the i-DIR method is the large wall-clock time required to solve the optimization problem, which can take up to 40 times the time required by optimized elastic DIR methods. This limitation may be alleviated by implementations that leverage the power of GPUs in DIR libraries [41]. In addition, due to the high computational demands, the current version of the i-DIR method has only been applied to 2D images. We remark that the motion and deformation analysis based on 2D images of the thorax constitutes an important limitation of this work in the biomechanical characterization of the lung. However, we also remark that under normal conditions, the dominant orientation of displacements in the lung is in the apico-basal direction [10], which is included in the sagittal images considered in this study. Future extensions should focus on DIR implementations for 3D CT thoracic images, based on which a complete biomechanical

study can be performed to fully understand the 3D nature of deformations in the lung. In addition, we note that due to the large level of strains experienced by the lung under full inspiration, the elastic energy component employed in the i-DIR formulation may not be suitable, as it corresponds to the elastic deformation energy for small strain levels [35]. To overcome this limitation, hyperelastic warping fomulations have been proposed which employ elastic energy terms that are compatible with large deformation [43]. The use of hyperelastic energy terms in the future versions of the i-DIR method constitutes a promising avenue of research.

**Author Contributions:** Conceptualization, D.E.H.; methodology, C.I.A. and D.E.H.; software, C.I.A.; validation, C.I.A. and D.E.H.; formal analysis, C.I.A. and D.E.H.; investigation, C.I.A. and D.E.H.; resources, D.E.H.; writing—original draft preparation, C.I.A. and D.E.H.; writing—review and editing, D.E.H.; visualization, C.I.A.; supervision, D.E.H.; project administration, D.E.H.; funding acquisition, D.E.H. Both authors have read and agreed to the published version of the manuscript.

**Funding:** This work received financial support from the Chilean National Agency for Research and Development (ANID) through grant FONDECYT Regular #1180832 awarded to D.E. and by the Millennium Science Initiative Program—NCN17_129. C.I.A. acknowledges the support of ANID through the CONICYT Doctoral fellowship.

**Institutional Review Board Statement:** Not applicable.

**Informed Consent Statement:** Not applicable.

**Data Availability Statement:** The data presented in this study are available on request from the corresponding author. The data are not publicly available due to privacy.

**Conflicts of Interest:** The authors declare no conflict of interest.

## Abbreviations

The following abbreviations are used in this manuscript:

| | |
|---|---|
| i-DIR | Inelastic deformable image registration |
| FEM | Finite element method |
| FFD | Free form deformation |
| TLC | Total lung capacity |
| FRC | Functional residual capacity |
| CT | Computed tomography |
| RSS | Residual sum of squared differences |
| TRE | Target registration error |

## Appendix A. Mathematical Definitions and Demonstrations

*Appendix A.1. Relation between the Rate of Plastic Strain and the Internal Variables*

In general, the relationship between $\dot{\varepsilon}^p$ and $q$ is not independent and is given by the *Prandtl-Reuss* flow rule:

$$\dot{\varepsilon}_{ij}^{p} = \dot{\bar{\varepsilon}}^{p}\left(\frac{3}{2}\frac{s_{ij}}{\bar{\sigma}}\right) \tag{A1}$$

where $\dot{\bar{\varepsilon}}^p$ stands for the evolution of the accumulated plastic strain and $\bar{\sigma}$ for the effective stress, which for the von Mises model takes the form

$$\bar{\sigma} = \sqrt{\frac{3}{2}s_{ij}s_{ij}}. \tag{A2}$$

For multi-axial plasticity we can rewrite the flow rule (20) as,

$$\dot{\varepsilon}_{ij}^{p} = \dot{\bar{\varepsilon}}^{p} M_{ij} \tag{A3}$$

where $M \equiv M_{ij} = \frac{3}{2}\frac{s_{ij}}{\bar{\sigma}}$ stands for the instantaneous direction of plastic flow. Similarly, the effective stress (A2) can be expressed as

$$\bar{\sigma} = \sigma_{ij}M_{ij}. \tag{A4}$$

Going a step back and clearing $\dot{\varepsilon}^p$ from (A1) we have,

$$\dot{\bar{\varepsilon}}^p = \sqrt{\frac{2}{3}\dot{\varepsilon}^p_{ij}\dot{\varepsilon}^p_{ij}} \tag{A5}$$

which in view of the plastic flow rule, is analogous to

$$\dot{\bar{\varepsilon}}^p = \dot{q} \tag{A6}$$

Then by integration of (A5) and again, assuming isotropic hardening, we have the following relation,

$$\bar{\varepsilon}^p = \int_0^t \sqrt{\frac{2}{3}\dot{\varepsilon}^p_{ij}\dot{\varepsilon}^p_{ij}}\, dt \equiv q \tag{A7}$$

Finally, we compute the driving forces for $q$ as,

$$y = -\frac{\partial A}{\partial q} = \frac{\partial W^e}{\partial \varepsilon^e_{ij}}M_{ij} - \frac{\partial W^p}{\partial q}(q) = \sigma_{ij}M_{ij} - \frac{\partial W^p}{\partial q}(q) \tag{A8}$$

$$y = \bar{\sigma} - \sigma_c. \tag{A9}$$

*Appendix A.2. Incremental Flow Rule Update*

Following an incremental flow rule of the type,

$$\varepsilon^p_{n+1} = \varepsilon^p_n + \Delta q M \tag{A10}$$

$$= \varepsilon^p_n + \Delta q \frac{3}{2}\frac{s_{n+1}}{\bar{\sigma}_{n+1}} \tag{A11}$$

let

$$g_n(\varepsilon_{n+1}, q_{n+1}) = A(\varepsilon_{n+1}, \varepsilon^p_{n+1}(q_{n+1}), q_{n+1}) - A_n + \Delta t \cdot \psi^*\left(\frac{|q_{n+1} - q_n|}{\Delta t}\right) \tag{A12}$$

Then we seek to minimize (A12) with respect to $q_{n+1}$, such that,

$$\frac{\partial g_n}{\partial q_{n+1}} = 0 \quad \Longrightarrow \quad \inf_{q_{n+1}} g_n(\varepsilon_{n+1}, q_{n+1}) \tag{A13}$$

we solve the above in a sub-differential way, such that

$$0 \in \frac{\partial A}{\partial q_{n+1}} + \partial\psi^*\left(\frac{|q_{n+1} - q_n|}{\Delta t}\right) \tag{A14}$$

$$0 \in \frac{\partial W^e}{\partial \varepsilon^e_{n+1}} \cdot \frac{\partial \varepsilon^e_{n+1}}{\partial q_{n+1}}(\varepsilon_{n+1} - \varepsilon^p_n - (q_{n+1} - q_n)M) + \frac{\partial W^p}{\partial q}(q_{n+1}) + \partial\psi^*\left(\frac{|q_{n+1} - q_n|}{\Delta t}\right) \tag{A15}$$

$$0 \in -\sigma_{n+1} \cdot M + \frac{\partial W^p}{\partial q}(q_n + \Delta q) + \partial\psi^*\left(\frac{|\Delta q|}{\Delta t}\right) \tag{A16}$$

$$0 \in -\bar{\sigma}_{n+1} + \frac{\partial W^p}{\partial q}(q_n + \Delta q) + \partial\psi^*\left(\frac{|\Delta q|}{\Delta t}\right) \tag{A17}$$

Recalling that the von Mises flow vector is purely deviatoric [36], we have that

$$
\begin{aligned}
\boldsymbol{s}_{n+1} &= 2\mu\boldsymbol{\varepsilon}_{n+1}^{e} \\
&= 2\mu(\boldsymbol{\varepsilon}_{n+1} - \boldsymbol{\varepsilon}_{n+1}^{p}) \\
&= 2\mu\left(\boldsymbol{\varepsilon}_{n+1} - \boldsymbol{\varepsilon}_{n}^{p} - \Delta q\frac{3}{2}\frac{\boldsymbol{s}_{n+1}}{\bar{\sigma}_{n+1}}\right) \\
&= 2\mu\left(\boldsymbol{\varepsilon}_{n+1} - \boldsymbol{\varepsilon}_{n}^{p}\right) - 2\mu\Delta q\frac{3}{2}\frac{\boldsymbol{s}_{n+1}}{\bar{\sigma}_{n+1}}
\end{aligned}
$$

$$
\boldsymbol{s}_{n+1} = \boldsymbol{s}_{n+1}^{pre} - 3\mu\Delta q\frac{\boldsymbol{s}_{n+1}}{\bar{\sigma}_{n+1}} \tag{A18}
$$

Since the predictive and updated deviatoric stress are co-linear ($\boldsymbol{s}_{n+1} \parallel \boldsymbol{s}_{n+1}^{pre}$), we can state that,

$$
\boldsymbol{M} = \frac{3}{2}\frac{\boldsymbol{s}_{n+1}}{\bar{\sigma}_{n+1}} = \frac{3}{2}\frac{\boldsymbol{s}_{n+1}^{pre}}{\bar{\sigma}_{n+1}^{pre}} \tag{A19}
$$

$$
\Longrightarrow \boldsymbol{s}_{n+1} = \frac{\boldsymbol{s}_{n+1}^{pre}}{\bar{\sigma}_{n+1}^{pre}}\bar{\sigma}_{n+1} \tag{A20}
$$

From the above we can re-write (A18) as,

$$
\boldsymbol{s}_{n+1} = \boldsymbol{s}_{n+1}^{pre} - 3\mu\Delta q\frac{\boldsymbol{s}_{n+1}^{pre}}{\bar{\sigma}_{n+1}^{pre}} \tag{A21}
$$

$$
\boldsymbol{s}_{n+1} = \left(1 - \frac{3\mu\Delta q}{\bar{\sigma}_{n+1}^{pre}}\right)\boldsymbol{s}_{n+1}^{pre} \tag{A22}
$$

Now replacing (A20) in (A22), we have,

$$
\frac{\boldsymbol{s}_{n+1}^{pre}}{\bar{\sigma}_{n+1}^{pre}}\bar{\sigma}_{n+1} = \left(1 - \frac{3\mu\Delta q}{\bar{\sigma}_{n+1}^{pre}}\right)\boldsymbol{s}_{n+1}^{pre} \tag{A23}
$$

$$
\Longrightarrow \bar{\sigma}_{n+1} = \bar{\sigma}_{n+1}^{pre} - 3\mu\Delta q \tag{A24}
$$

Going a step back to (A17) we can explicitly define,

$$
\bar{\sigma}_{n+1}^{pre} - 3\mu\Delta q \in \frac{\partial W^{p}}{\partial q}(q_{n} + \Delta q) + \partial\psi^{*}\left(\frac{|\Delta q|}{\Delta t}\right) \tag{A25}
$$

Finally, assuming that $W^{p} = \frac{1}{2}Hq_{n+1}^{2}$ and $\psi^{*} = \sigma_{y}|\Delta q|$, we can rewrite (A25) as:

$$
\bar{\sigma}_{n+1}^{pre} - 3\mu\Delta q - H(q_{n} + \Delta q) - \sigma_{y} = 0 \tag{A26}
$$

which eventually delivers,

$$
\Delta q = \frac{\bar{\sigma}_{n+1}^{pre} - Hq_{n} - \sigma_{y}}{3\mu + H} \tag{A27}
$$

*Appendix A.3. Return Mapping Algorithm*

The solution of $\Delta q$ in (A27) involves a two mutually exclusive steps:

(i) An elastic predictor, such that,

$$
\begin{aligned}
\Delta q &= 0 \\
q_{n+1}^{pre} &= q_n \\
\bar{\sigma}_{n+1}^{pre} &= \{2\mu(\varepsilon_{n+1} - \varepsilon_{n+1}^p(q_{n+1}^{pre})) + \lambda \, \mathrm{trace}(\varepsilon_{n+1} - \varepsilon_{n+1}^p(q_{n+1}^{pre}))I\} \cdot M \\
&= \sqrt{\tfrac{3}{2} s_{n+1}^{pre} \cdot s_{n+1}^{pre}}
\end{aligned}
\tag{A28}
$$

replacing in (A26), we have

$$
\bar{\sigma}_{n+1}^{pre} - Hq_n - [-\sigma_y, \sigma_y] = 0 \tag{A29}
$$

$$
\implies \begin{cases} \bar{\sigma}_{n+1}^{pre} &\leq \quad Hq_n + \sigma_y \\ \bar{\sigma}_{n+1}^{pre} &\geq \quad -Hq_n - \sigma_y \end{cases} \tag{A30}
$$

and

(ii) A plastic corrector, where we have two possible cases:

(a) if the elastic trial lies within the elastic domain

$$
\Phi(\bar{\sigma}_{n+1}^{pre}) \leq 0 \qquad \implies \qquad \bar{\sigma}_{n+1}^{pre} \in [-\sigma_y, \sigma_y] \tag{A31}
$$

there is no plastic evolution within the time interval $(t_n, t_{n+1})$, and therefore we update our variables:

$$
(\cdot)_{n+1} = (\cdot)_{n+1}^{pre} \tag{A32}
$$

and

(b) otherwise, we have plastic flow (or elasto-plastic evolution). By a traditional Newton-Raphson linearization we solve the following

$$
\bar{\sigma}_{n+1}^{pre} - 3\mu\Delta q - H(q_n + \Delta q) - \sigma_y = 0 \tag{A33}
$$

and then we update the following variables at $t_{n+1}$,

$$
\varepsilon_{n+1}^p = \varepsilon_n^p + \Delta q \frac{3}{2} \frac{s_{n+1}}{\bar{\sigma}_{n+1}} \tag{A34}
$$

$$
q_{n+1} = q_n + \Delta q \tag{A35}
$$

$$
\varepsilon_{n+1}^e = \varepsilon_{n+1} - \varepsilon_{n+1}^p \tag{A36}
$$

$$
\sigma_{n+1} = \lambda \, \mathrm{trace}(\varepsilon_{n+1}^e)I + 2\mu\varepsilon_{n+1}^e \tag{A37}
$$

$$
\mathbf{D}_{n+1}^{ep} = 2\mu\left(1 - \frac{3\mu\Delta q}{\bar{\sigma}_{n+1}^{pre}}\right)\mathbf{I}_d + 6\mu^2\left(\frac{\Delta q}{\bar{\sigma}_{n+1}^{pre}} - \frac{1}{3\mu + H}\right)\bar{\mathbf{M}}_{n+1} \otimes \bar{\mathbf{M}}_{n+1} + K\mathbf{I} \otimes \mathbf{I} \tag{A38}
$$

where $K$ is the bulk modulus, $\bar{\mathbf{M}}_{n+1} \equiv \sqrt{\tfrac{2}{3}}\mathbf{M}_{n+1} = \frac{s_{n+1}^{pre}}{\|s_{n+1}^{pre}\|}$ is the unit plastic flow vector and $\mathbf{I}_d$ is the fourth order deviatoric projection tensor defined as,

$$
\mathbf{I}_d \equiv \mathbf{I}_S - \frac{1}{3}\mathbf{I} \otimes \mathbf{I} \tag{A39}
$$

with $\mathbf{I}_S = \frac{1}{2}(\delta_{ik}\delta_{jl} + \delta_{il}\delta_{jk})$ as the fourth order symmetric identity tensor.

*Appendix A.4. Effective Incremental Energy*

The **effective** incremental energy can be defined as follows,

$$
W_n(\varepsilon) = \inf_{q_{n+1}} g_n(\varepsilon, q_{n+1}) = g_n(\varepsilon, q_{n+1}^*(\varepsilon)) \tag{A40}
$$

where,

$$q^*_{n+1}(\varepsilon) \in \operatorname{argmin} g_n(\varepsilon, q_{n+1}) \tag{A41}$$

such that,

$$\frac{\partial g_n}{\partial q_{n+1}}(\varepsilon, q^*_{n+1}(\varepsilon)) = 0 \tag{A42}$$

Conveniently know the effective potential depends solely in $\varepsilon = \varepsilon_{n+1}$. Following this definition, we compute the first and second derivative of $W_n$, where,

$$DW_n(\varepsilon) = \frac{\partial g_n}{\partial \varepsilon}(\varepsilon, q^*_{n+1}(\varepsilon)) \tag{A43}$$

$$= \frac{\partial g_n}{\partial \varepsilon}(\varepsilon, q^*_{n+1}(\varepsilon)) + \frac{\partial g_n}{\partial q_{n+1}}(\varepsilon, q^*_{n+1}(\varepsilon)) \cdot \frac{\partial q^*_{n+1}}{\partial \varepsilon}(\varepsilon) \tag{A44}$$

but since $\frac{\partial g_n}{\partial q_{n+1}}(\varepsilon, q^*_{n+1}(\varepsilon)) = 0$, we have that for a fully implicit scheme,

$$DW_n(\varepsilon) = \frac{\partial g_n}{\partial \varepsilon}(\varepsilon, q^*_{n+1}(\varepsilon)) \equiv \sigma_{n+1} \tag{A45}$$

where $\sigma_{n+1}$ are the stresses at $t = t_{n+1}$.

Then we compute,

$$D^2 W_n(\varepsilon) = \frac{\partial^2 g_n}{\partial \varepsilon \partial \varepsilon}(\varepsilon, q^*_{n+1}(\varepsilon)) + \frac{\partial^2 g_n}{\partial \varepsilon \partial q_{n+1}}(\varepsilon, q^*_{n+1}(\varepsilon)) \cdot \frac{\partial q^*_{n+1}}{\partial \varepsilon}(\varepsilon) \tag{A46}$$

where we can redefine $\frac{\partial q^*_{n+1}}{\partial \varepsilon}(\varepsilon)$ deriving (A42) by $\frac{\partial}{\partial \varepsilon}$, such that,

$$\frac{\partial^2 g_n}{\partial q_{n+1} \partial \varepsilon}(\varepsilon, q^*_{n+1}(\varepsilon)) + \frac{\partial^2 g_n}{\partial q_{n+1} \partial q_{n+1}}(\varepsilon, q^*_{n+1}(\varepsilon)) \cdot \frac{\partial q^*_{n+1}}{\partial \varepsilon}(\varepsilon) = 0 \tag{A47}$$

$$\implies \frac{\partial q^*_{n+1}}{\partial \varepsilon}(\varepsilon) = -\left\{ \frac{\partial^2 g_n}{\partial q_{n+1} \partial q_{n+1}}(\varepsilon, q^*_{n+1}(\varepsilon)) \right\}^{-1} \cdot \frac{\partial^2 g_n}{\partial q_{n+1} \partial \varepsilon}(\varepsilon, q^*_{n+1}(\varepsilon)) \tag{A48}$$

replacing in (A46), we have that for a fully implicit scheme,

$$D^2 W_n(\varepsilon) = \frac{\partial^2 g_n}{\partial \varepsilon \partial \varepsilon}(\varepsilon, q^*_{n+1}(\varepsilon)) - \frac{\partial^2 g_n}{\partial \varepsilon \partial q_{n+1}}(\varepsilon, q^*_{n+1}(\varepsilon)) \cdot \left\{ \frac{\partial^2 g_n}{\partial q_{n+1} \partial q_{n+1}}(\varepsilon, q^*_{n+1}(\varepsilon)) \right\}^{-1} \cdot \frac{\partial^2 g_n}{\partial q_{n+1} \partial \varepsilon}(\varepsilon, q^*_{n+1}(\varepsilon)) \tag{A49}$$

$$D^2 W_n(\varepsilon) \equiv \mathbf{D}^{ep}_{n+1} \tag{A50}$$

where $\mathbf{D}^{ep}_{n+1}$ is known as the consistent tangent modulus at $t = t_{n+1}$.

*Appendix A.5. Finite-Element Discretization of the I-Dir Formulation*

Let:

$$\boldsymbol{u} \approx u^h_i := \sum_{A=1}^{n} N_A u_{iA} = \mathbf{N}(x)\mathbf{u} \tag{A51}$$

$$\boldsymbol{v} \approx v^h_j := \sum_{A=1}^{n} N_A v_{jA} = \mathbf{N}(x)\mathbf{v} \tag{A52}$$

$$\varepsilon(\nabla \boldsymbol{u}) \approx \varepsilon(\nabla \boldsymbol{u}^h) := \sum_{A=1}^{n} B_A u_{iA} = \mathbf{B}(x)\mathbf{u} \tag{A53}$$

$$\varepsilon(\nabla \boldsymbol{v}) \approx \varepsilon(\nabla \boldsymbol{v}^h) := \sum_{A=1}^{n} B_A v_{jA} = \mathbf{B}(x)\mathbf{v} \tag{A54}$$

where $N_A$ are the shape functions, **N** is the matrix of global shape functions, and **B** is the global strain-displacement matrix.

$$\mathbf{N} = \begin{bmatrix} N_1 & 0 & 0 & N_2 & \cdots & N_i & 0 & 0 \\ 0 & N_1 & 0 & 0 & \cdots & 0 & N_i & 0 \\ 0 & 0 & N_1 & 0 & \cdots & 0 & 0 & N_i \end{bmatrix} \tag{A55}$$

$$\mathbf{B} = \begin{bmatrix} \mathbf{B}_1 & \mathbf{B}_2 & \mathbf{B}_3 & \cdots \mathbf{B}_i \end{bmatrix} \tag{A56}$$

$$\mathbf{B}_i = \begin{bmatrix} \frac{\partial N_i}{\partial x} & 0 & 0 \\ 0 & \frac{\partial N_i}{\partial y} & 0 \\ 0 & 0 & \frac{\partial N_i}{\partial z} \\ \frac{\partial N_i}{\partial y} & \frac{\partial N_i}{\partial x} & 0 \\ 0 & \frac{\partial N_i}{\partial z} & \frac{\partial N_i}{\partial y} \\ \frac{\partial N_i}{\partial z} & 0 & \frac{\partial N_i}{\partial x} \end{bmatrix} \tag{A57}$$

Substituting the approximations (A51)–(A54) into the linear variational problem (40) we obtain the linear system of equations defined in (41), where the tangent matrix and residual vector are defined as

$$\mathbf{K}_n := \alpha \int_{\Omega^h} \mathbf{N}^T \left\{ \nabla T(\boldsymbol{u}_n^h) \otimes \nabla T(\boldsymbol{u}_n^h) + (T(\boldsymbol{u}_n^h) - R)\nabla\nabla T(\boldsymbol{u}_n^h) \right\} \mathbf{N} + \int_{\Omega^h} \mathbf{B}^T \mathbf{D}_{n+1}^{ep}(\varepsilon(\boldsymbol{u}_n^h)) \mathbf{B}, \tag{A58}$$

$$\mathbf{F}_n := \alpha \int_{\Omega^h} \mathbf{N}^T (T(\boldsymbol{u}_n^h) - R)\nabla T(\boldsymbol{u}_n^h) + \int_{\Omega^h} \mathbf{B}^T \sigma_{n+1}(\varepsilon(\boldsymbol{u}_n^h)), \tag{A59}$$

which are constructed by numerically evaluating the element expressions and assembling their contributions into the global matrix and vector using standard finite-element techniques, see, e.g., [40].

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
