# Peer review of "Inelastic Deformable Image Registration (i-DIR): Capturing Sliding Motion through Automatic Detection of Discontinuities"

_mathematics, doi:10.3390/math9010097_

Round 1
Reviewer 1 Report
This manuscript presents a novel inelastic (i) deformable image registration (DIR: i-DIR) method to detect sliding surfaces and handle sliding discontinuities in computed tomography (CT) images. Unlike other methods, i-DIR is truly predictive because it does not require a priori information about sliding surfaces. The originality in the i-DIR formalism lies in its utilization of the von Mises plasticity model from continuous mechanics for imaging purposes. In essence, this formalism incorporates an inelastic energy term to detect regions of high shearing deformation related to sliding. I find this theoretical development as really original and significant. The i-DIR method is compared with other imagining methods in their performances on 2-D images from a synthetic dataset and CT images of the lung. Overall, the i-DIR method performs better as determined by various metrics. The use of 2-D images and of only one case of anatomical/medical relevance (CT images of the lung) is somewhat preliminary but provides enough evidence of the potential of i-DIR, which can be further validated with future tests.
Scientific background, medical relevance, methodology and results are satisfactorily discussed. In particular, the mathematical formalism of i-DIR from the von Mises theory is explained clearly and completely in the main text and appendices. The English of the manuscript is very good. The length of the manuscript is appropriate. As stated previously, the i-DIR formalisms is indeed innovative and shows promising results. I think that this manuscript will be valuable for the Mathematics readership and I am glad to recommend it for publication in its present form.
Author Response
Comments from Reviewer 1
This manuscript presents a novel inelastic (i) deformable image registration (DIR: i-DIR) method to detect sliding surfaces and handle sliding discontinuities in computed tomography (CT) images. Unlike other methods, i-DIR is truly predictive because it does not require a priori information about sliding surfaces. The originality in the i-DIR formalism lies in its utilization of the von Mises plasticity model from continuous mechanics for imaging purposes. In essence, this formalism incorporates an inelastic energy term to detect regions of high shearing deformation related to sliding. I find this theoretical development as really original and significant. The i-DIR method is compared with other imagining methods in their performances on 2-D images from a synthetic dataset and CT images of the lung. Overall, the i-DIR method performs better as determined by various metrics. The use of 2-D images and of only one case of anatomical/medical relevance (CT images of the lung) is somewhat preliminary but provides enough evidence of the potential of i-DIR, which can be further validated with future tests.
Scientific background, medical relevance, methodology and results are satisfactorily discussed. In particular, the mathematical formalism of i-DIR from the von Mises theory is explained clearly and completely in the main text and appendices. The English of the manuscript is very good. The length of the manuscript is appropriate. As stated previously, the i-DIR formalisms is indeed innovative and shows promising results. I think that this manuscript will be valuable for the Mathematics readership and I am glad to recommend it for publication in its present form.
Answer: We thank the reviewer for the positive comments and strong support.
Reviewer 2 Report
The topic is very interesting. The study is well organized. However, the paper is disorganized, too long in many sections, thus being confusing.
Many grammar errors along the paper.
It should be completely revised before considering it suitable for publication.
Author Response
Comments from Reviewer 2
The topic is very interesting. The study is well organized. However, the paper is disorganized, too long in many sections, thus being confusing.Many grammar errors along the paper. It should be completely revised before considering it suitable for publication.
Answer: We thank the reviewer for the support and comments. The manuscript has been restructured with shorter subsections to give more clarity to the reader and has been thoroughly revised. Grammar has been reviewed accordingly.
Reviewer 3 Report
Dear Authors,
The topic of the image registration is very interesting especially when it covers the problem of so-called large deformations. Please find my remarks below.
In the Introduction you focus on a few DIR techniques directly connected to the problem of sliding. In my opinion the most common DIR techniques should be specified earlier, including works for example by John Ashburner (J Ashburner: A fast diffeomorphic image registration algorithm. NeuroImage 2007, 38(1):95-113). At least some classification of DIR methods should be provided to the reader.
Starting form the Introduction through the whole paper, the problem of the organ sliding is connected strictly with a lungs. To my knowledge it is present in many parts of the human body. It might be not clear for the reader why you focus only on this organ.
To assess the performance of the method on anatomical images, you considered sagital planes of a CT thorax. The question arises, why this data-set is considered only as a 2D image. The respiratory movement does not act in just one direction. This simplification may lead to some errors, I see some in Fig. 9 inside the thorax for all methods, and this may be the cause. Please comment on that.
The measure RSS in my opinion is better than the TRE. The TRE reflect the quality of registration in the most significant places (e.g. rib landmarks) which is of course desirable, but this measure acts only locally. Placing landmarks inside the thorax in different locations may change in a large extend the results. The i-DIR method behaves better outside the thorax (see Table 3, TRE for rib landmarks) but not inside - values of RSS and Fig 9. indicate it.
A minor final note: I would not consider DIR as an image-analysis tool, rather as a method or as a process.
Author Response
Comments from Reviewer 3
The topic of the image registration is very interesting especially when it covers the problem of so-called large deformations. Please find my remarks below.
In the Introduction you focus on a few DIR techniques directly connected to the problem of sliding. In my opinion the most common DIR techniques should be specified earlier, including works for example by John Ashburner (J Ashburner: A fast diffeomorphic image registration algorithm. NeuroImage 2007, 38(1):95-113). At least some classification of DIR methods should be provided to the reader.
Answer: We thank the reviewer for the support, and for this specific suggestion. We have expanded the introduction to describe how DIR methods can be classified according to the deformation model assumed, which includes the Diffeomorphic DIR method, please see the revised manuscript.
Starting form the Introduction through the whole paper, the problem of the organ sliding is connected strictly with a lungs. To my knowledge it is present in many parts of the human body. It might be not clear for the reader why you focus only on this organ.
Answer: Following this comment, we have included other examples of sliding that occurs inside the human body, please see the revised manuscript. Our focus on the lung is explained by our interest in understanding the biomechanics and deformation of lung parenchyma, which can be highly distorted when using standard DIR methods, as stated in the introduction.
To assess the performance of the method on anatomical images, you considered sagital planes of a CT thorax. The question arises, why this data-set is considered only as a 2D image. The respiratory movement does not act in just one direction. This simplification may lead to some errors, I see some in Fig. 9 inside the thorax for all methods, and this may be the cause. Please comment on that.
Answer: This is an important limitation of our work, which is now described in detail in the discussion’s section of the revised manuscript. Differences that arise in Fig 9 can be effectively related to the out-of-plane displacement that occurs between end of expiration and end of inspiration images. Despite these differences, the largest displacements in the lung have been observed in the apico-basal direction, which is included in the sagittal images considered in this study, and which may give a reasonable first order approximation.
The measure RSS in my opinion is better than the TRE. The TRE reflect the quality of registration in the most significant places (e.g. rib landmarks) which is of course desirable, but this measure acts only locally. Placing landmarks inside the thorax in different locations may change in a large extend the results. The i-DIR method behaves better outside the thorax (see Table 3, TRE for rib landmarks) but not inside - values of RSS and Fig 9. indicate it.
Answer: We thank the reviewer for this important comment. In effect, the quality of the registration inside the lungs does not substantially changes between the i-DIR and the other elastic DIR methods. However, it is important to note that in this work we pursue the automatic detection of sliding in thorax CT images, and i-DIR shows a clear advantage in this regard when compared to traditional elastic and enhanced DIR methods. These conclusions have been expanded in the discussion section of the revised manuscript.
A minor final note: I would not consider DIR as an image-analysis tool, rather as a method or as a process.
Answer: Thank you for this suggestion; ‘tool’ has been changed to ‘method’ in the manuscript.
Reviewer 4 Report
Research on the inelastic deformable image registration (DIR) with automatic detection of discontinuities using capturing sliding motion is valuable. The results achieved provides the new mathematical base on the DIR application for the image registration. The novelty of the work is the method ability to detect automatically sliding surfaces of the registered images and its capability of handling sliding discontinuities.
Publication would benefit if the authors would shortly discuss the novelty of the proposed method with respect to other unsupervised methods in DIR field.
The proposed method is thoroughly described – this is a strong part of the study.
The results of the proposed method are presented shortly and are limited only to two test cases. The performance metrics shows improvement, and the future research field is outlined with the authors understanding of the current approach limitations.
For the minor improvement would be enough if the authors:
· shortly discuss the novelty/ disadvantages/advantages of the proposed method with respect to other unsupervised methods in DIR field.
Author Response
Comments from Reviewer 4
Research on the inelastic deformable image registration (DIR) with automatic detection of discontinuities using capturing sliding motion is valuable. The results achieved provides the new mathematical base on the DIR application for the image registration. The novelty of the work is the method ability to detect automatically sliding surfaces of the registered images and its capability of handling sliding discontinuities.
The proposed method is thoroughly described – this is a strong part of the study.
The results of the proposed method are presented shortly and are limited only to two test cases. The performance metrics shows improvement, and the future research field is outlined with the authors understanding of the current approach limitations.
Answer: We thank the reviewer for this positive comment and strong support to our work.
For the minor improvement would be enough if the authors:
- shortly discuss the novelty/ disadvantages/advantages of the proposed method with respect to other unsupervised methods in DIR field. Publication would benefit if the authors would shortly discuss the novelty of the proposed method with respect to other unsupervised methods in DIR field.
Answer: Thank you for this comment. We have included a discussion about other unsupervised DIR methods, and how they compare to our i-DIR formulation in the discussion section of the manuscript.
Round 2
Reviewer 2 Report
The Authors made great efforts in ameliorating the paper according to reviewers' comments.
The paper now merits publication on Mathematics